# *BRCA1* and *BRCA2* Gene Mutations and Lung Cancer Risk: A Meta-Analysis

**DOI:** 10.3390/medicina56050212

**Published:** 2020-04-27

**Authors:** Yen-Chien Lee, Yang-Cheng Lee, Chung-Yi Li, Yen-Ling Lee, Bae-Ling Chen

**Affiliations:** 1Department of Oncology, Tainan Hospital, Ministry of Health and Welfare, Tainan 700, Taiwan; yenpig8291@gmail.com; 2Department of Internal Medicine, National Cheng Kung University Hospital, College, Tainan 700, Taiwan; 3Tainan Municipal Hospital, Tainan 700, Taiwan; leeyangjason2@gmail.com; 4Department of Public Health, College of Medicine, National Cheng Kung University, Tainan 700, Taiwan; cyli99@mail.ncku.edu.tw; 5Department of Public Health, College of Public Health, China Medical University, Taichung 404, Taiwan; 6Department of Healthcare Administration, College of Medical and Health Science, Asia University, Taichung 413, Taiwan; 7College of Intelligence, National Taichung University of Science and Technology, Taichung 404, Taiwan; chenbl@nutc.edu.tw

**Keywords:** lung cancer, *BRCA1*, *BRCA2*, meta-analysis

## Abstract

*Background and objective: BRCA1* and *BRCA2* are associated with many cancer types in addition to hereditary breast and ovarian cancers. However, their relation to lung cancer remains to be explored. *Materials and Methods:* Observation studies were systematically reviewed to explore the association of *BRCA1* or *BRCA2* with lung cancer. PubMed, MEDLINE [EBSCOhost], and relevant articles published up to 7 January 2020 were searched. Odd ratio (OR), standardized morbidity rate (SMR), and cancer-specific standardized incidence ratios (SIRs) were pooled together as relative risk (RR) estimates (95% confidence interval [CI], 0.66–1.40). *Results:* Thirteen studies were included for analysis. Results showed that the RR of *BRCA2* is 0.76 (95% CI, 0.48–1.19), the overall RR is 0.96 (95% CI, 0.66–1.40), and that of *BRCA1* is 0.66 (95% CI, 0.41–1.05), indicating that it was not associated with lung cancer. *Conclusion:* With the limitation of the retrospective study design and severe heterogeneity, these results inform clinicians and relevant families that *BRCA1* and *BRCA2* mutation carriers have no increased risk of lung cancer.

## 1. Introduction

Genetic testing has been widely used since the discovery of the breast cancer-associated genes *BRCA1* in 1994 and *BRCA2* in 1995 [1]. “USPSTF Calls for More *BRCA* Screening” is a recent US Preventive Services Task Force statement reiterating the importance of screening for *BRCA1/2* mutations, especially for those with a personal history of certain cancer types and certain ancestries [2]. In January 2018, the first poly (ADP-ribose) polymerase (PARP) inhibitor was approved by the Food and Drug Administration (FDA) for *BRCA* mutation metastatic breast cancer [3]. Considering the flourishing of the genetic test and correlated target therapy, the association of the *BRCA* gene to other cancer types must be clarified. *BRCA1* and *BRCA2* encode large proteins and bear minimal resemblance to one another. These tumor suppressor genes play an important role in the DNA double-strain repair system and are widely expressed during the S and G2 phases in different tissues [4]. *BRCA1* plays a proximal and extensive role in the cellular response to double-strand breaks, and *BRCA2* controls the *RAD51* recombinase essential for the repair of double-strand breaks by homologous DNA sequences (HR) [4]. In addition to causing hereditary breast and ovarian cancer syndrome, these genes also increase other cancer risks, including pancreatic and prostate cancer. However, their roles in lung cancer remain controversial. Several studies [5,6] showed that lung cancer risks are increased, whereas others [7,8] claimed that the effect is irrelevant. 

Whether *BRCA* genes are drivers of mutations for lung cancer remains unknown and thus must be confirmed by epidemiology studies. Considering the conflicting results, this study aimed to determine the relationship between lung cancer and *BRCA* mutation via overall and stratified meta-analyses based on current epidemiology research.

## 2. Materials and Methods

### 2.1. Search Strategy and Data Abstraction

PubMed and MEDLINE [EBSCOhost] databases were systematically searched for relevant articles published up to 7 January 2020 by using the term “*BRCA*” without language restriction. Additional search methods included a manual review of the reference lists of relevant studies. Inclusion criteria were as follows: (1) Published with extractable information on lung cancer incidence, (2) the participants had *BRCA1* or *BRCA2* mutation, and (3) the control groups involved patients without mutation or the general population. The effects of the *BRCA* gene on the occurrence of lung cancer was analyzed with proper control. Studies regarding *BRCA1* or *BRCA2* with control groups were selected. Abstract or posters were not selected because their quality is difficult to evaluate. When the same patient population was used, the work with the highest patient number was selected. Cohort studies with ascertained *BRCA* mutation carriers and cohort studies involving pedigree analysis were analyzed together, and the ascertained ones were selected for analysis. 

Preferred Reporting Items for Systematic Reviews and Meta-Analyses statement was followed for data extraction. Two reviewers (YCL and YLL) independently examined the title and the abstract of the publications according to the search strategy. The full texts of all potentially relevant publications were retrieved. Information extracted from each study included the publication year, the name of the first author, the trial type, the patient number, observed cases, control cases, odd ratio (OR), standardized morbidity rate (SMR) and cancer-specific standardized incidence ratios (SIRs), and relative risk (RR).

The Newcastle–Ottawa scale was used for quality assessment in the cohort study and consists of eight items for a total of 9 points as follows: Representativeness of the cohort, selection of control cohort, ascertainment of exposure, demonstration that outcome of interest not present at the start of study, comparability (two points, study controls for age and one another factor), assessment of outcome, follow-up long enough for outcomes to occur, and adequacy of follow-up of cohorts.

### 2.2. Statistics

OR, SMR, and SIR were treated as equivalent measures of risks and pooled together as RR estimates. An estimate from Figure was calculated while exact number could not be obtained from the study. The same control group was used for *BRCA1* and *BRCA2*, and the control number was divided into two and run up to the integer. RRs were pooled across studies by using inverse-variance weighted DerSimonian–Laird random-effect models to allow for between-study heterogeneity. *I*^2^ statistics were used for between-study heterogeneity. *I*^2^ is the proportion of the total variation in the estimated effects for each study due to heterogeneity between studies. Analyses were conducted in Stata (version 12.0; Stata Corp, College Station, TX, USA). Two-sided *p* < 0.05 was statistically significant. 

## 3. Results

From a review of 3498 total titles or abstracts, 16 full articles were retrieved. Additionally, 23 full articles were obtained through the manual review of the reference lists of relevant articles (Figure 1). From these 39 articles, 26 were excluded because of the following reasons: (1) Repeat cohorts in 4 articles, (2) no association with lung cancer in 19 articles, (3) 2 articles with no control groups, and (4) one article did not focus on the *BRCA* gene. Finally, 13 articles were selected (Figure 1). Three studies were conducted in multiple countries [5,8,9]. Among the 10 remaining studies, three were conducted in the United States [7,10,11], four in Europe (including Sweden [12], the Netherlands [13], the United Kingdom [14], and Italy [6]), one in South Korea [15], one in Canada [16], and one in Israel [17]. Two reports were obtained from the same cohort MD Anderson [7,10], and the other two were possibly from the same Toronto cohort [5,16]. The studies with a large group of patients were selected for the group analysis [10] (Figure 2). Considering the different subgroup representatives, each study was used in subgroup analysis. The observed and expected number was estimated from the figure in one study due to not provide a precise number in the article. [17].

Five cohort studies involved ascertained *BRCA* mutation carriers [5,7,8,12,17]. The remaining cohort involved the pedigree to expand the cohort number by calculating the risk of being a carrier. Nine studies compared the cancer statistics of the general population [5,7,8,9,10,12,13,14,17], two compared the same proband patients with *BRCA* without mutation [6,11], and the remaining two compared the same institute for patients with cancer [15,16] (Table 1). Three studies grouped *BRCA1* with *BRCA2* for analysis [5,6,11] (Table 1), and one work focused on men [17].

The qualities of the cohort studies were evaluated according to the Newcastle–Ottawa Scale with the highest possible total score of 9. The distributions of total scores for the 13 studies are as follows: 7 (three studies), 6 (four studies), 5 (one study), and 4 (five studies) (Table 2). The first score, which is the “representativeness of the exposed cohort”, was scored as positive, indicating that the study was considered truly or somewhat representative of the *BRCA* mutation carriers. The second score, which is the “selection of the non-exposed cohort”, was scored as positive, indicating that the general population or no *BRCA* mutation was selected. The third score, which is the “ascertainment of exposure”, relates to the measurement of *BRCA* initially at the start of the study. The fourth score, which is the “demonstration that the outcome of interest was not present at the start of the study”, is scored as positive when lung cancer was not presented initially. The fifth score, which is the “comparability of the cohorts on the basis of design or analysis”, was scored according to whether the analysis set the initial age and/or an additional factor as a control variable. The sixth score, which is the “assessment of outcome”, was scored positively when the procedure of lung cancer confirmation was described. The seventh score is “was follow-up long enough for outcomes to occur”. A median follow-up of greater than 5 years would be adequate. The eighth and final score, which is the “adequacy of follow-up of cohorts”, was scored positively when the follow-up was complete or the subjects lost to follow-up were less than 20%. Only four studies confirmed lung cancer diagnosis by chart reviews [5,7,8,13]. The median follow-up was not reported. Only one work reported follow-up adequacy [12].

Five cohort studies ascertained *BRCA* mutation carriers [5,7,8,12,17]. All the five studies compared the population-specific incidence rate. One study grouped *BRCA1* with *BRCA2* and focused on male patients [17]. When the two genes were pooled together, *BRCA1* was not associated with increased lung cancer risk with RR of 0.66 (95% CI, 0.41–1.05), *BRCA2* had an RR of 0.76 (95% CI, 0.48–1.19), and the overall RR is 0.96 (95% CI, 0.66–1.40, Figure 2). Sensitivity analysis was conducted for reclassifying the different study designs. Considering that Johannsson O joined the Breast Cancer Linkage Consortium, these three studies might have been duplicated [8,9,12]. Breast Cancer Linkage Consortium [8,9] used a large population, which was duplicated in the desired subgroup analysis. Three cohort studies yielded RR of 0.87 (95% CI, 0.35–2.14) for *BRCA1* carriers, three studies reported the RR of 1.57 for *BRCA2* carriers (95% CI, 0.76–3.25), and one grouping *BRCA1* with *BRCA2* indicated the RR of 1.43 (95% CI, 0.2–10.15, Figure 3). Four cohort studies involved pedigree analysis [9,10,13,14] and compared the population-specific incidence rate. A two-pedigree cohort yielded an RR of 0.43 (95% CI, 0.32–0.58) for *BRCA1* carriers, four pedigree-cohort reported an RR of 0.56 for *BRCA2* carriers (95% CI, 0.38–0.83), and two grouping *BRCA1* with *BRCA2* indicated an RR of 2.34 (95% CI, 0.73–7.52). Two studies compared *BRCA* mutation carriers with the branches of the family belonging to the same proband *BRCA* patients [6,11]. The studies grouping *BRCA1* with *BRCA2* reported an RR of 2.34 (95% CI, 0.73–7.52). The remaining two studies grouping *BRCA1* with *BRCA2* compared the patients with cancer in the same institute [15,16] (Figure 3). Considering their similarity in their control groups (population-specific incidence rate), the cohort studies were grouped with cohort pedigree studies for analysis while excluding those that might overlap. The RR for *BRCA1* was 0.61 (95% CI, 0.37–0.99), and that for *BRCA2* was 0.74 (95% CI, 0.46–1.18, Figure 4).

## 4. Discussion

This study showed that families with *BRCA1* or *BRCA2* have no increased risks for lung cancer on the basis of current available studies. Many diversities are present regarding the reporting of age. Some studies reported age in categories, others in mean or median, and some did not report at all (Table 1). Most studies involved white people, and some involved Westerners. Only one study was from Korea. Therefore, subgroup analysis according to age or ethnicity could not be performed. A high heterogeneity was observed in the study designs, and *I*^2^ tests with clinical significance, except for the subgroup analysis with cohort studies only (Figure 3). The sensitivity analysis of different study designs yielded the same results (Figure 3). Studies with ascertained *BRCA* mutation showed the absence of an increased lung cancer risk (Figure 3). Some works included pedigree analysis by family members with a 50% prior probability of being a carrier regardless of their known carrier status [13], and others included all first-degree relatives [14]. Some reports included first- and second-degree relatives for analysis [10]. For comparison, some studies used the general population, including the same family of proband and other cancer patients in the same hospital (Table 1). After these individuals were excluded in the case control studies, the result showed that *BRCA* mutation did not increase the risk of lung cancer (Figure 4). 

Many studies were derived from the same cohort (BCLC cohort [8,9,12], MD Anderson cohort [7,10], and Ontario cohort [5,16]). A previous study grouped duplicate cohorts for analysis [18]. Brose MS et al. identified their *BRCA1* mutation carriers either on the basis of direct genetic testing or as presumed carriers [19]. Oh M et al. [18] pointed out that although 96 out of 147 of the families included in Brose et al. were duplicated, the sensitivity test was not substantially affected. They also grouped case-control and pedigree studies. However, another study by Ford et al. [20] was duplicated without being mentioned. In the present work, all possible duplicated studies were excluded for analysis. Subgroup analysis was also conducted as the different control groups might have different implications (Figure 3). *BRCA1* and *BRCA2* studies requires many resources, which make only a few centers available for data collection and analysis. Hence, duplicates were often included. Some studies expanded the cohort by including pedigree. Only works that included a direct gene test cohort study and were selected for our meta-analysis revealed that lung cancer incidence was not increased in *BRCA1* or *BRCA2* carriers (Figure 3).

*BRCA2* regulates the availability and activity of RAD51, which has a catalytic activity central to recombination between homologous DNA sequences; meanwhile, *BRCA1* links DNA damage sensing [4]. Though their different roles, these proteins work collectively to protect the genome from double-strand DNA damage during DNA replication [21]. Carriers show *BRCA1* or *BRCA2* loss because of germline and somatic mutations or promoter hypermethylation [22]. Increased risks for prostate and pancreatic cancer occur in families with the *BRCA2* mutation [13,21,23]. Colorectal cancer risk increases in *BRCA1* but not in *BRCA2* mutation [18]. *BRCA* might not be a driver mutation of lung cancer. If the *BRCA* carrier does not increase the risk of lung cancer, PARP inhibitors might not be efficient for *BRCA* carriers with lung cancer, e.g., olaparib [24]. 

One study reported a decreased lung cancer risk for non-smoking individuals compared with that for smoking people because of the awareness of lifestyle factors that increase cancer risk [13]. Smoking increases the risk of any cancer in patients with the *BRCA* mutation [25]. *BRCA* carriers reduce their smoking behavior after knowing their genetic test results [26] and this might be the reason for the lack of an increase of lung cancer risk compared with the higher prevalence of smoking behavior of the control group. However, the smoking pattern among the included studies is unknown. None of the studies adjusted for smoking status. 

Our studies have some limitations. Data on outcomes, such as lung cancer, have unknown accuracy. Four studies claimed that the analyses were restricted to the use of verified cancer [5,7,14,17], but their definition of lung cancer diagnosis in the *BRCA* family is unknown. Whether the definition is obtained from a pathology report or recalled from family relatives remains unclear. Follow-up completion also remains unknown, and heterogeneity must be acknowledged. The ascertained *BRCA* mutation carriers also differ. Some studies used a direct sequence, whereas others applied risk estimation.

## 5. Conclusions

Our meta-analysis showed that individuals who are *BRCA1* or *BRCA2* carriers do not have an increased risk of lung cancer.

## Figures and Tables

**Figure 1 medicina-56-00212-f001:**
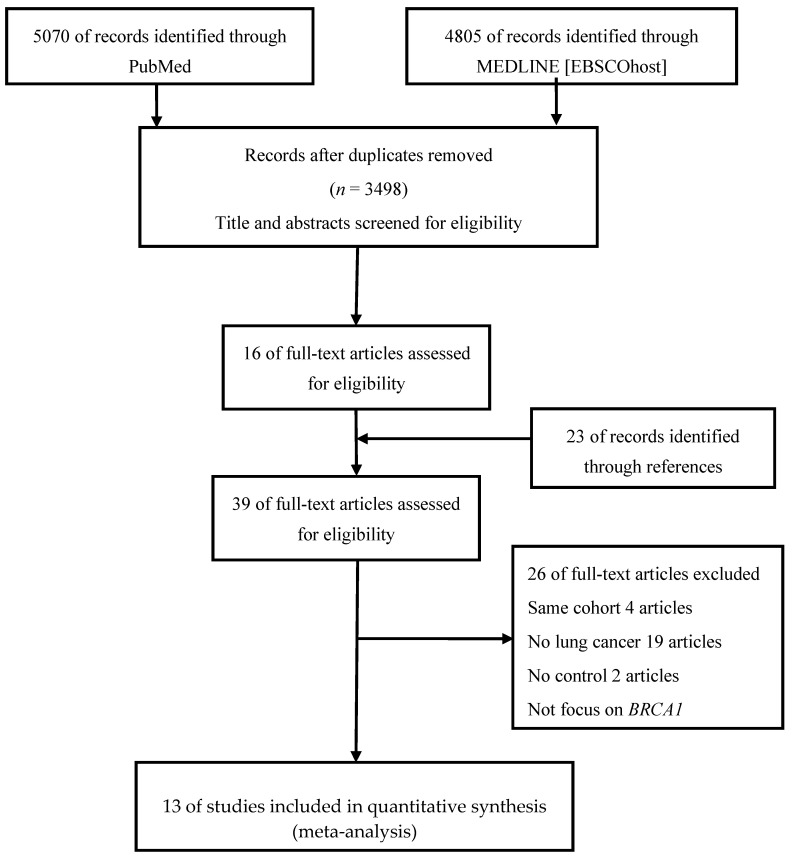
Selection of studies in the meta-analysis.

**Figure 2 medicina-56-00212-f002:**
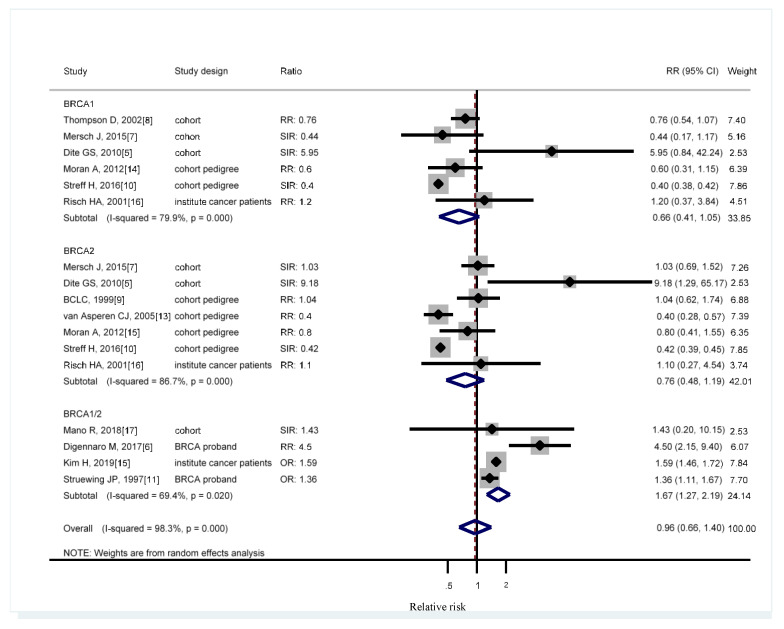
Meta-analysis of *BRCA*
*gene* mutation and lung cancer incidence. CI: confidence interval, RR: relative risk, SIR: cancer specific standardized incidence ratio, OR: odds ratio, SMR: standardized morbidity rate. Solid diamonds denote ratio point estimate from each study, open diamonds represent pooled overall results, and dashed line denotes overall pooled point.

**Figure 3 medicina-56-00212-f003:**
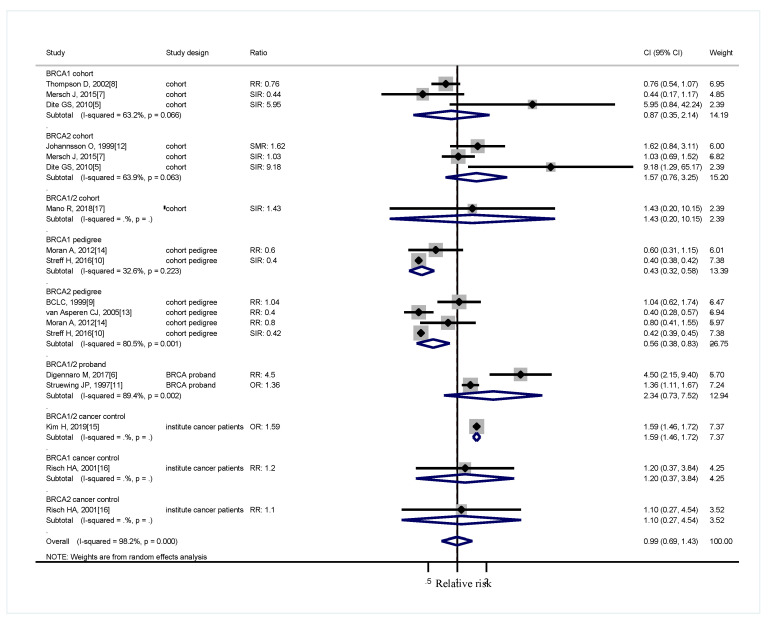
Sensitivity analysis of different study designs. CI: confidence interval, RR: relative risk, SIR: cancer specific standardized incidence ratio, OR: odds ratio, SMR: standardized morbidity rate. Solid diamonds denote ratio point estimate from each study, open diamonds represent pooled overall results, and dashed line denotes overall pooled point.

**Figure 4 medicina-56-00212-f004:**
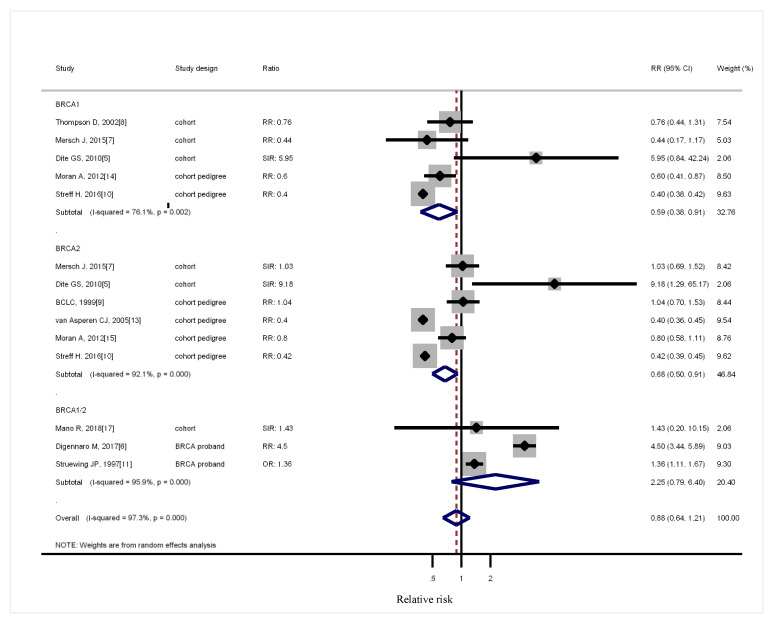
Sensitivities analysis of cohort studies. CI: confidence interval, RR: relative risk, SIR: cancer specific standardized incidence ratio, OR: odds ratio, SMR: standardized morbidity rate. Solid diamonds denote ratio point estimate from each study, open diamonds represent pooled overall results, and dashed line denotes overall pooled point.

**Table 1 medicina-56-00212-t001:** Summary of the baseline characteristics of the included studies.

Cohort Studies	Study Design	Participants *BRCA* (M,F) * Age **	Observed Cases +	Controls ++	Control Cases	Reported Estimated (95% CI)
Cohort studies with ascertained *BRCA* mutation carriers						
Johannsson O et al., 1999 [12]	South Swedish healthcare Cohort	*BRCA1* 1086 (547,539) *BRCA2* 684 (366,318) Age (NA)	6 3	2.80 1.85	Sweden Cancer Registries (1958–1995)	SMR = 2.15 (0.79–4.67) SMR = 1.62 (0.33−4.73)
Thompson D et al., 2002 [8]	BCLC (Breast Cancer Linkage Consortium), Western Europe, the US and Canada	*BRCA1* 2245 (NA) Age by stratification	5	13.05	Cancer Incidence in Five Continents	RR = 0.76 (0.54–1.07)
Dite GS et al., 2010 [5]	Caucasian, SanFrancisco (USA), Ontario (Canada), Melbourne and Sydney (Australia)	*BRCA1* 25 (NA) *BRCA2* 17 (NA) Age (NA)	1 1	0.17 0.11	1925-1985 Connecticut, USA; 1965–2001 Ontario, Canada; 1983-2001 Australia	SIR = 5.95 (0.84–42.21) SIR = 9.18 (1.29–65.20)
Mersch J et al., 2015 [7]	Cohort (MD Anderson)	*BRCA1* 613 (NA) *BRCA2* 459 (NA) Mean age 49.3 ± 12.76	2 5	4.547 4.867	US Cancer Statistics (1999–2010)	SIR = 0.440 (0.049–1.588) SIR = 1.027 (0.331–2.398)
Mano R et al., 2018 [17]	Israel male	*BRCA1* 117 *BRCA2* 79 median age 49	1	0.7	Age-adjusted cancer incidence, Israeli Jewish male population in Israel-National Cancer register	Not available
Cohort studies involving pedigree analysis						
BCLC, 1999 [9]	BCLC (Breast Cancer Linkage Consortium), Western Europe, the US and Canada	*BRCA*2 3728 (NA) Age (NA)	9	11.43	Cancer Incidence in Five Continents	RR = 1.04 (0.62–1.73)
van Asperen CJ et al., 2005 [13]	Cohort, Netherlands	*BRCA*2 1811 (NA) Ascertain by 50% prior probability of being a carrier Age (NA)	30	40.4	Eindhoven Cancer Registry to 1990 and Netherlands Cancer Registry from 1990	RR = 0.4 (0.3–0.6)
Moran A et al., 2012 [14]	Cohort United Kingdom	*BRCA1* 1815 (715,1100) *BRCA2* 1526 (595,931) Frist degree relatives Age (NA)	8.2 10.9	14.2 13.2	North West of England (1975–2005)	RR = 0.6 (0.3–1.1) RR = 0.8 (0.4–1.5)
Streff H et al., 2016 [10]	Cohort (MD Anderson)	*BRCA1* 5237 (2401,2836) *BRCA2* 3795 (1802,1993) First, 2nd degree relatives Age (NA)	33 30	83.8 70.8	U.S Cancer Statistics (1999–2011)	SIR = 0.40 (0.27–0.55) SIR = 0.42 (0.29–0.61)
Cohort studies with special control						
Struewing JP et al., 1997 [11]	Cohort, control, branches of the family belonging to the same proband *BRCA* patients	*BRCA 1/2* 114 (NA) *BRCA(-)* 4759 (NA) Age (in categories)	11	337	Relatives of cases with no mutations	NA
Digennaro M et al., 2017 [6]	Cohort, Italy Control, branches of the family belonging to the same proband *BRCA* patient	*BRCA 1/2* 1156 (NA) *BRCA*(-) 1062 (NA) Age mean *BRCA1/2* 65.9, *BRCA (-)* 69.1	38	9	2004–2008 consultation in a single center	RR = 4.5 (2.15–9.38)
Risch HA et al., 2001 [16]	Cohort, Ontario, Canada; from ovary cancer related family	*BRCA1* 39 (NA) *BRCA2* 21 (NA) *BRCA*(-)455 (NA) Age (in categories)	4.5% 4.2%	3.7%	Relatives of cases with no mutations	RR = 1.2 (0.38–3.9) RR = 1.1(0.27–4.6)
Kim H et al., 2019 [15]	Cohort, Korea Control, history of cancer other than breast or ovarian cancer	*BRCA 1/2* 377 (6, 371) *BRCA*(-) 2178 (13, 2165) Age median 40	33 109	109	Breast cancer patients in a single institute	OR = 1.586 (1.057–2.380)

* *BRCA*(M,F): *BRCA* mutation (male number, female number); ** age: mean or mean age of the study group; +: lung cancer number, ++: controls: lung cancer number of the control group, *BRCA1/2: BRCA1* or *BRCA2* mutation, *BRCA*(-): no *BRCA* mutation, NA: none available, CI: confidence interval, RR: relative risk, SIR: cancer specific standardized incidence ratio, OR: odds ratio, SMR: standardized morbidity rate.

**Table 2 medicina-56-00212-t002:** Methodologic quality of studies, based on the Newcastle-Ottawa scale (*N* = 13).

Studies	Representativeness of the *BRCA* Gene	Selection of the Non-*BRCA* or General Population	Ascertainment of *BRCA*	Demonstration That Lung Cancer Presented	Study Controls for Initial Age and/or for an Additional Factor	Assessment of Outcome	Was Median Follow-Up 5 Years or More?	Adequacy of Follow-Up (>80%)	Total
Johannsson O [12]	★	★	★	★	★★	-	-	★	7
Thompson D [8]	★	★	★	★	-,-	-	-	-	4
Dite GS [5]	★	★	★	★	★-	★	-	-	5
Mersch J [7]	★	★	★	★	★★	★	-	-	7
Mano R [17]	★	★	★	★	★-	★	-	-	6
BCLC [9]	★	★	★	★	-,-	-	-	-	4
van Asperen CJ [13]	★	★	★	★	★★	-	-	-	6
Moran A [14]	★	★	★	★	★ ★	★	-	-	7
Streff H [10]	★	★	★	★	★★	-	-	-	6
Digennaro M [6]	★	★	★	★	★★	-	-	-	6
Kim H [15]	★	★	★	★	-,-	-	-	-	4
Struewing JP [11]	★	★	★	★	-,-	-	-	-	4
Risch HA [16]	★	★	★	★	-,-	-	-	-	4

The first score, which is the “representativeness of the exposed cohort”, was scored as positive, indicating that the study was considered truly or somewhat representative of the *BRCA* mutation carriers. The second score, which is the “selection of the non-exposed cohort”, was scored as positive, indicating that the general population or no BRCA mutation was selected. The third score, which is the “ascertainment of exposure”, relates to the measurement of *BRCA* initially at the start of study. The fourth score, which is the “demonstration that the outcome of interest was not present at start of the study”, is scored as positive when lung cancer was not presented initially. The fifth score, which is the “comparability of the cohorts on the basis of design or analysis”, was scored according to whether the analysis set the initial age and/or an additional factor as a control variable. The sixth score, which is the “assessment of outcome”, was scored positively when the procedure of lung cancer confirmation was described. The seventh score is “was follow-up long enough for outcomes to occur”. A median follow-up of greater than 5 years would be adequate. The eighth and final score, which is the “adequacy of follow-up of cohorts”, was scored positively when the follow-up was complete or the subjects lost to follow-up were less than 20%. ★: score one point, -: no point of the score.

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
