# Peer review of "BRCA1 and BRCA2 Gene Mutations and Lung Cancer Risk: A Meta-Analysis"

_medicina, 2020, doi:10.3390/medicina56050212_

Round 1
Reviewer 1 Report
In this paper the authors performed a meta analysis of 13 papers to study the association of BRCA mutations and lung cancer risk. They didn’t find any association.
The article is moderately interesting, but quite badly written and should go through some major changes to make it more clearly written and easier to understand.
In the introduction, line 30 – authors didn’t mention that before breast cancer, PARP inhibitors were already used for treating BRCA positive ovarian cancer. There are also clinical studies for PARP inhibitors for prostate and some other cancers. This data should be included.
Line 63 in the Materials and Methods – Abbreviations should be explained
Lines 74,75 – I2 is written incorrectly
Figure 1 – pubmed and medline are underlined
Figure 2 – very unclear and hard to read, it should be written using larger letters in text and smaller line spacing.
Table 1 – references are not correctly numbered
Table 2 – references are not correctly numbered
Table 2 – minuses are not the same size, is there any reason for that?
Figures 2, 3 and 4, and Tables 1 and 2 should have the references in the same order. It would help the reader to compare different aspects. Maybe in order of Fig 2, since it makes the most sense to order the cohorts that way. The authors explain the reorganization in Fig 3, but not very clearly.
Several studies are missing in some tables and figures, and are included in others. The authors should explain which study is omitted where and why in figure/table legends.
Acknowledgments is left from instructions.
Through the text gene and protein names are sometimes italicized when a proteins are described, and sometimes not where genes are in question. It should be corrected.
Author Response
Reviewer #1:
In this paper the authors performed a meta analysis of 13 papers to study the association of BRCA mutations and lung cancer risk. They didn’t find any association.
- The article is moderately interesting, but quite badly written and should go through some major changes to make it more clearly written and easier to understand.
We have rewritten it with the help of a native English speaker.
- In the introduction, line 30 – authors didn’t mention that before breast cancer, PARP inhibitors were already used for treating BRCA positive ovarian cancer. There are also clinical studies for PARP inhibitors for prostate and some other cancers. This data should be included.
We believed that PARP inhibitors have been approved for other cancers besides breast cancer as well as other sporadic clinical data. However, we want to focus our study aim to BRCA gene and lung cancer incidence. Only mention about PARP inhibitor due to if BRCA gene were a driver mutation for lung cancer, PARP inhibitor might be a treatment choice, but the study results showed otherwise.
If the reviewer thought it is really important to put emphasis on PARP related studies, please tell us again. We will add more information on the following revision.
- Line 63 in the Materials and Methods – Abbreviations should be explained
These abbreviations have been already mention in the abstract section. Please see abstract section, line 16-18. “Odd ratio (OR), standardized morbidity rate (SMR), and Cancer specific standardized incidence ratios (SIR) were pooled together as relative risk (RR) estimates.”
- Lines 74,75 – I2 is written incorrectly
We have corrected it to I2. Thank you very much.
- Figure 1 – pubmed and medline are underlined
We have deleted the underlined. Thank you.
- Figure 2 – very unclear and hard to read, it should be written using larger letters in text and smaller line spacing.
We have changed the size of the letter from 4 to 5. And had adjusted them to make them more readable.
- Table 1 – references are not correctly numbered
We have corrected renumber them. Thank you very much.
- Table 2 – references are not correctly numbered
We have corrected renumber them. Thank you very much.
- Table 2 – minuses are not the same size, is there any reason for that?
We have changed them to the same size. Thank you very much.
- Figures 2, 3 and 4, and Tables 1 and 2 should have the references in the same order. It would help the reader to compare different aspects. Maybe in order of Fig 2, since it makes the most sense to order the cohorts that way. The authors explain the reorganization in Fig 3, but not very clearly.
We have reorder them all. Thank you very much.
- Several studies are missing in some tables and figures, and are included in others. The authors should explain which study is omitted where and why in figure/table legends.
There are 13 studies in total. Some are cohort studies and some are case control studies. Also, some study only focus on BRCA1, some on BRCA2 and some grouping them together. That’s why different figure with different studies. However, table 1 and 2 should be equal total of 13 studies.
- Acknowledgments is left from instructions.
We have filled it up. Thank you.
- Through the text gene and protein names are sometimes italicized when a proteins are described, and sometimes not where genes are in question. It should be corrected.
We have corrected them. Thank you.
Reviewer 2 Report
The authors systematically reviewed observation studies and carried out meta-analysis and suggested that BRCA1 and BRCA2 gene mutations were not associated with lung cancer risk. This study is an interesting approach but contains limitations in diagnosis method and gene mutation uncertainties. The authors should explain the gender and age distribution in the cohorts and discuss the results from these points and also from racial differences.
-Fonts, capitalization and spacing are not unified in the Figures, Tables and the text. -Grammar and mis-spelling should be corrected throughout the text.
-Gene names should be converted to italic throughout the text.
-Page 1, line 26: reference(s) should be necessary.
-Page 1, line 31: reference(s) should be necessary.
-Page 1, line 33: The spelling of “Theses” should be corrected.
-Page 1, line 35: should be corrected to “phases”.
-Page 2, line 63, the abbreviations, “OR, SMIR, SIR and RR” needs explanation.
- Page 2, line 63, references should be necessary.
-Table 1: The meanings of “observed cases” and “control” are not clear and the numbers should be checked.
-Table 1: The meanings of BRCA(+) and BRCA(-) should be explained.
-Table 2: The meanings of one star, two stars, and bars should be explained.
-Figs. 3 and 4: Figure components including the dotted line should be explained for wide readers outside of the statistics.
-Fig. 2: The sizes of the font are too small and should be enlarged; some characters are not clearly presented.
Author Response
Reviewer #2
The authors systematically reviewed observation studies and carried out meta-analysis and suggested that BRCA1 and BRCA2 gene mutations were not associated with lung cancer risk. This study is an interesting approach but contains limitations in diagnosis method and gene mutation uncertainties. The authors should explain the gender and age distribution in the cohorts and discuss the results from these points and also from racial differences.
Thank you for the valuable comments. Different studies presented in different ways. Some reported age in categories, some in mean or median while others reported none. We have added gender and age distribution in table 1. We have added below in the discussion section line 24. “ There were lots of diversities regarding reporting of age. Some studies reported age in categories while others in mean or median or not reported at all (Table 1). Regarding to ethnicity, most belongs to white and most studies were from Westerns. Only one study was from Korea. We couldn’t perform subgroup analysis according to age or ethnicity.”
-Fonts, capitalization and spacing are not unified in the Figures, Tables and the text.
We have corrected them. Thank you.
-Grammar and mis-spelling should be corrected throughout the text.
We have rewritten it with the help of a native English speaker.
-Gene names should be converted to italic throughout the text.
We have corrected them. Thank you.
-Page 1, line 26: reference(s) should be necessary.
We have inserted reference [1].
-Page 1, line 31: reference(s) should be necessary.
We have inserted reference [3].
-Page 1, line 33: The spelling of “Theses” should be corrected.
We have corrected it. Thank you.
-Page 1, line 35: should be corrected to “phases”.
We have corrected it. Thank you.
-Page 2, line 63, the abbreviations, “OR, SMIR, SIR and RR” needs explanation.
These abbreviations have been already mention in the abstract section. Please see abstract section, line 16-18. “Odd ratio (OR), standardized morbidity rate (SMR), and Cancer specific standardized incidence ratios (SIR) were pooled together as relative risk (RR) estimates.”
- Page 2, line 63, references should be necessary.
Here are the methods. “The information that was extracted from each study included the publication year, the name of the first author, the trial type, the patient number, observed cases, control cases, as well as OR, SMR, SIR and RR. “ So there would be no references.
-Table 1: The meanings of “observed cases” and “control” are not clear and the numbers should be checked.
We have added notification below table 1. +: lung cancer number,++: controls: lung cancer number of the control group.
-Table 1: The meanings of BRCA(+) and BRCA(-) should be explained.
We have change BRCA(+) into BRCA1/2 and had put notification below to showed that BRCA1/2: BRCA1 or 2 mutation, BRCA(-): no BRCA mutation,
-Table 2: The meanings of one star, two stars, and bars should be explained.
We have added below to make them more readable.
“The first score, “representativeness of the exposed cohort”, is scored as positive where the study was considered truly or somewhat representative of the BRCA mutation carriers. The second score, “selection of the non-exposed cohort”, is scored as positive if the general population or no BRCA mutation were selected. The third score, “ascertainment of exposure”, relates to measurement of BRCA in our analysis. The fourth score, “demonstration that the outcome of interest was not present at start of the study”, is scored as positive as lung cancer was diagnosis. The fifth score, “comparability of the cohorts on the basis of design or analysis”, was scored according to whether the analysis controlled for initial age and/or for an additional factor. The sixth score, “assessment of outcome”, was scored positively if procedure of lung cancer confirmation was described. The seventh score is “was follow-up long enough for outcomes to occur”. It was decided that a median follow-up of greater than 5 years would be adequate. The eighth and final score, “adequacy of follow up of cohorts” was scored positively if the follow-up was complete or the subjects lost to follow up were less than 20%. «: score one point, -: no point of the score”
-Figs. 3 and 4: Figure components including the dotted line should be explained for wide readers outside of the statistics.
We have added “Solid diamonds denote ratio point estimate from each study, open diamonds represent pooled overall results, and dashed line denotes overall pooled point.” below Figs.2, 3, 4.
-Fig. 2: The sizes of the font are too small and should be enlarged; some characters are not clearly presented.
We have changed the size of the letter from 4 to 5. And had adjusted them to make them more readable.
Reviewer 3 Report
In the paper there is nothing really new. Very well known that BRCA1/2 genes are related with hereditary breast and ovarian cancer, but there is no relation with lung cancer.
Selected articles sample size is unclear and it is difficult to estimate the size of the samples evaluated. The question is, what are the sampling criteria?
There are no conclusions about how many samples are need to be taken for RR to be significant.
There are no meaningful arguments for what is missing from the research.
Gene names must be written in italic.
The first figure requires correction.
There is no table 2 legend,, it is not clear what one or two stars mean, why the lengths of the dashes differ.
And very poor picture quality (2, 3 and 4).
Author Response
In the paper there is nothing really new. Very well known that BRCA1/2 genes are related with hereditary breast and ovarian cancer, but there is no relation with lung cancer.
In Kim H et al, 2019 et al studies, the OR=1.586 (1.057-2.380) showed that BRCA mutation was associated with lung cancer. Also, in Dite GS et al, 2010 studies showed SIR=5.95 (0.84-42.21) and SIR=9.18 (1.29-65.20) for BRCA1 and BRCA2 respectively. We can never tell until we group all the current evidence together to know the final result.
Selected articles sample size is unclear and it is difficult to estimate the size of the samples evaluated. The question is, what are the sampling criteria?
The criteria has been stated in the methods section. We included all the currently known studies related to BRCA and lung cancer.
“We systematically searched Pubmed and Medlines [EBSCOhost] for relevant articles published up to Jan 7, 2020, with the terms, “BRCA” without language restriction. Additional search methods included manual review of the reference lists of relevant studies. Studies were included if: (1) They were published with extractable information on lung cancer incidence; (2) Patients included in the study with BRCA1 or BRCA2 mutation; (3) Control groups were patients without mutation or general population. The goal of the analysis was to examine the effects of BRCA gene on the occurrence of lung cancer with proper control. Studies regarding BRCA1 or BRCA2 with control groups were selected. Abstract or posters were not selected owing to hard to evaluate quality. When the same patient population was used, study with more patient number was selected. While the cohort studies with ascertained BRCA mutation carriers and cohort studies involving pedigree analysis were analyzed together, the ascertained ones were selected into analysis.”
There are no conclusions about how many samples are need to be taken for RR to be significant.
It depends on disease incidence and study population. One can never tell.
There are no meaningful arguments for what is missing from the research.
We can adjusted them if this reviewer make the statements clearer. Thank you.
Gene names must be written in italic.
We have corrected it. Thank you.
The first figure requires correction.
We have corrected it.
There is no table 2 legend,, it is not clear what one or two stars mean, why the lengths of the dashes differ.
The legend has been stated in the result section. We have pasted it below table 2 to make it more readable.
And very poor picture quality (2, 3 and 4).
We have changed the sized of figure 2 and tried to make figure 3 and 4 clearer. Thank you.
Round 2
Reviewer 1 Report
Thank you for making the changes. The article is much easier to read and understand now.
Some minor points:
In the footnote for Table 2 BRCA is not in italics.
There is something wrong with the legend of Figure 3.
Author Response
Thank you for making the changes. The article is much easier to read and understand now.
Thank you for the helpful comments.
Some minor points:
In the footnote for Table 2 BRCA is not in italics.
We have corrected them. Thank you.
There is something wrong with the legend of Figure 3.
We have deleted the sentences. Thank you.
Reviewer 2 Report
The authors have improved the manuscript, but following points still need revision.
-Page 2, line 63, the abbreviations, “OR, SMIR, SIR and RR” needs explanation. Generally, even if abbreviations are already mentioned in the abstract section, you need to define again in the text when first appeared.
-Table 1: The description of “characteristics age of the study group” is not understandable.
-Table 2, notification: Gene names should be in italic. Line breaks should be corrected.
-Table 2, notification: The description of “to the measurement of BRCA in our analysis” is not understandable. The final sentence is missing a period.
-Fig. 3: “dashed line” is not present and should be added.
-Figs. 2-4: The bottom scales of “0.5, 1, 2” should be checked and explained in the legend.
-Acknowledgments: The description of “We thank the reviewers and the editors for helpful comments and suggestions.” should be removed but should include, for example, the comments for supported colleagues or organization for this study.
-The English should be corrected including following parts.
Page 1, line 37: “DNA double strain repair system”
Page 10, line 13: Two pedigree cohort
Page 12, line 75: “BRCA might not be a driver mutation”
Fig. 1: “13of”. “Meta-analysis” (should be in small letters)
Author Response
Comments and Suggestions for Authors
The authors have improved the manuscript, but following points still need revision.
-Page 2, line 63, the abbreviations, “OR, SMIR, SIR and RR” needs explanation. Generally, even if abbreviations are already mentioned in the abstract section, you need to define again in the text when first appeared.
We have added it to page 2 of 14, lines 66 to 68.
-Table 1: The description of “characteristics age of the study group” is not understandable.
We have change it into “age: mean or mean age of the study group” to make it more readable.
-Table 2, notification: Gene names should be in italic. Line breaks should be corrected.
We have change the name in italic.
-Table 2, notification: The description of “to the measurement of BRCA in our analysis” is not understandable. The final sentence is missing a period.
We have change the sentence into “to the measurement of BRCA initially at the start of the study” and the missing period has been added. Thank you.
-Fig. 3: “dashed line” is not present and should be added.
It has presented but close to the solid line which make it hard to see.
-Figs. 2-4: The bottom scales of “0.5, 1, 2” should be checked and explained in the legend.
We have added relative risk bellows.
-Acknowledgments: The description of “We thank the reviewers and the editors for helpful comments and suggestions.” should be removed but should include, for example, the comments for supported colleagues or organization for this study.
We have removed this sentence.
-The English should be corrected including following parts.
Page 1, line 37: “DNA double strain repair system”
We have corrected it into “DNA double-strain repair system”.
Page 10, line 13: Two pedigree cohort
Can the reviewer specify it, so we can improve it. Thank you.
Page 12, line 75: “BRCA might not be a driver mutation”
Can the reviewer specify it, so we can improve it. Thank you.
Fig. 1: “13of”. “Meta-analysis” (should be in small letters)
We have corrected them. Thank you.
Reviewer 3 Report
Still there is no conclusions about how many samples needed to have significant RR. In the In Kim H et al, 2019 et al studies there is over two thousands, probably you will get significant RR. The same with BRCA1 and BRCA2. Very well known that BRCA2 mutations are more harmful to patient compare to BRCA1.
In the abstract, line 17, should be BRCA1, but not BRCA 1. Line 47 should be via, but not via.
In figure 1, should be BRCA1, but not BRCA 1.
In the table 2, in the legend should be BRCA gene name written in italic.
Page 10, line 28.. "estimate" something is missing?
Figure 4 font to large.
Author Response
Still there is no conclusions about how many samples needed to have significant RR. In the In Kim H et al, 2019 et al studies there is over two thousands, probably you will get significant RR. The same with BRCA1 and BRCA2. Very well known that BRCA2 mutations are more harmful to patient compare to BRCA1.
In the abstract, line 17, should be BRCA1, but not BRCA 1. Line 47 should be via, but not via.
We have changed them. Thank you.
In figure 1, should be BRCA1, but not BRCA 1.
We have changed them. Thank you.
In the table 2, in the legend should be BRCA gene name written in italic.
We have changed them. Thank you.
Page 10, line 28.. "estimate" something is missing?
We have deleted the sentences. Thank you.
Figure 4 font to large.
Can the reviewer specify it, so we can improve it. Thank you.